# Characterization of aerosols generated by high-power electronic nicotine delivery systems (ENDS): Influence of atomizer, temperature and PG:VG ratios

Seyed Ahmad Reza Dibaji[1☯], Berk Oktem[1☯], Lee Williamson[2☯], Jenna DuMond[2], Todd Cecil[2], Jimin P. Kim[2], Samanthi Wickramasekara[1], Matthew Myers[1], Suvajyoti Guha[1]*

1 Office of Science and Engineering Laboratories, Silver Spring, MD, United States of America, 2 Center for Tobacco Products, Office of Science, Silver Spring, MD, United States of America

☯ These authors contributed equally to this work.
* Suvajyoti.Guha@fda.hhs.gov

**Data Availability Statement:** All relevant data are within the paper and its Supporting Information files.

## Abstract

The aerosol characteristics of electronic nicotine delivery systems (ENDS) are important parameters in predicting health outcomes since parameters such as aerosol particle size correlate strongly to aerosol delivery and deposition efficiency. However, many studies to date do not account for aerosol aging, which may affect the measurement of ultra-fine particles that typically coagulate or agglomerate during puff development. To reduce aerosol aging, we herein present a unique instrumentation method that combines a) positive pressure ENDS activation and sample collection, b) minimization of both sample tubing length and dilution factors, and c) a high-resolution, electrical low-pressure impactor. This novel approach was applied to systematically investigate the effects of coil design, coil temperature, and propylene glycol to vegetable glycerol ratios on aerosol characteristics including aerosol mass generation, aerosol count generation, and the mass and count size distributions for a high-powered ENDS. Aerosol count measurements revealed high concentrations of ultra-fine particles compared to fine and coarse particles at 200˚C, while aerosol mass measurements showed an increase in the overall aerosol mass of fine and coarse particles with increases in temperature and decreases in propylene glycol content. These results provide a better understanding on how various ENDS design parameters affect aerosol characteristics and highlight the need for further research to identify the design parameters that most impact ultra-fine particle generation.

## Introduction

Electronic nicotine delivery systems (ENDS) have rapidly risen in popularity and usage in recent years [1, 2]. With this explosive growth in usage, particularly among the youth population, there is a major effort to understand the possible health risks of ENDS such as the

**Funding:** Center for Tobacco Products under award number PA-DPS-001-16.

**Competing interests:** The authors have declared that no competing interests exist.

harmful effects of aerosol constituents. In their simplest form, ENDS use a resistively heated coil to heat an e-liquid mixture of propylene glycol (PG), vegetable glycerol (VG), water, and nicotine, along with smaller amounts of flavoring compounds. Due to this simplicity and the lack of combustion processes, there is a general perceived reduced risk to the user and often ENDS have been presented as a potentially less risky option than smoking [3], even though the health risks of the products have not been well demonstrated [4]. The primary health risk results from possible aerosol constituents produced during heating and inhaled into the user's lungs. Heating of e-liquid has been shown to produce chemical byproducts and free radicals via thermal degradation, some of which are not found in cigarettes, while heavy metals have been found in e-cigarette aerosols, likely originating from the coil or soldered joints [5, 6]. Moreover, ENDS generate liquid droplets suspended in a mixture of vapor and air, while cigarettes produce smoke that is composed of fine solids and semi-volatile particles suspended in air [7]. Additionally, the likelihood of nanoparticle generation from resistively heated wires is well-supported by numerous works [8]. With the emergence of this new ENDS technology that has been rapidly evolving over the past decade, a corresponding vast body of research is needed to examine the health and safety risks of ENDS from various perspectives and fields of discipline [9].

The aerosol characteristics of ENDS, such as particle size, and the concentration of fine and ultra-fine particulate matter, are crucial parameters in predicting health outcomes. Specifically, the particle size distribution and aerosol mass generated by an ENDS determines deposition patterns and delivery efficiency within the user's respiratory tract, which has been demonstrated experimentally previously [10] and via mathematical simulation [11]. Moreover, identifying correlations between various ENDS design parameters and aerosol characteristics may present valuable tools in evaluating the design and health risks of ENDS. Research studies suggest that aerosol characteristics including aerosol mass, particle mass distributions, and count size distributions are affected by the coil operating temperature [12], the ratio of propylene glycol and vegetable glycerin (PG:VG) within the e-liquid [13–15] and the composition, design, and placement of the coil [16]. The coil type and number also play a role in the aerosol mass per puff generated by ENDS [17]. However, the impact of ENDS design parameters on the aerosol generated has largely been overlooked in most studies.

In addition to the limited research on ENDS aerosols, the dynamic nature of ENDS aerosols may often confound results. ENDS aerosols may quickly evaporate, absorb water, or coagulate due to the hygroscopic properties of PG and VG and the relative volatility of PG and water [18], resulting in experimental artifacts and high sampling variability. Thus, the accurate measurement of ENDS aerosol particle size requires near real-time capability to capture these dynamic changes during puff development. A survey of literature reveals that only rapid analyzers such as Electrical Low Pressure Impactors (ELPI), Fast Mobility Particle Spectrometer (FMPS) or Differential Mobility Spectrometer (DMS500) are sufficiently sensitive to detect particles smaller than 0.1 μm.

Fig 1 qualitatively demonstrates the particle size distribution as reported by studies from 2013–2019 [5, 12, 19–26]. These studies use different analyzers with some that were incapable of real time measurements, which may have impacted aerosol aging due to the time taken for analysis relative to the time scale of other events such as coagulation or growth of aerosols. Additionally, different aerosol dilutions, tubing lengths, sample collection methods, and ENDS types were employed in these studies. These factors may have contributed to the particle size distribution differences reported in these studies. Although the data is diverse, most studies reported a bimodal distribution with particle sizes that were either below 50 nm or above 200 nm being most common. However, the diversity in these findings show that near real-time instrumentation and better controlled experimental design are needed to produce

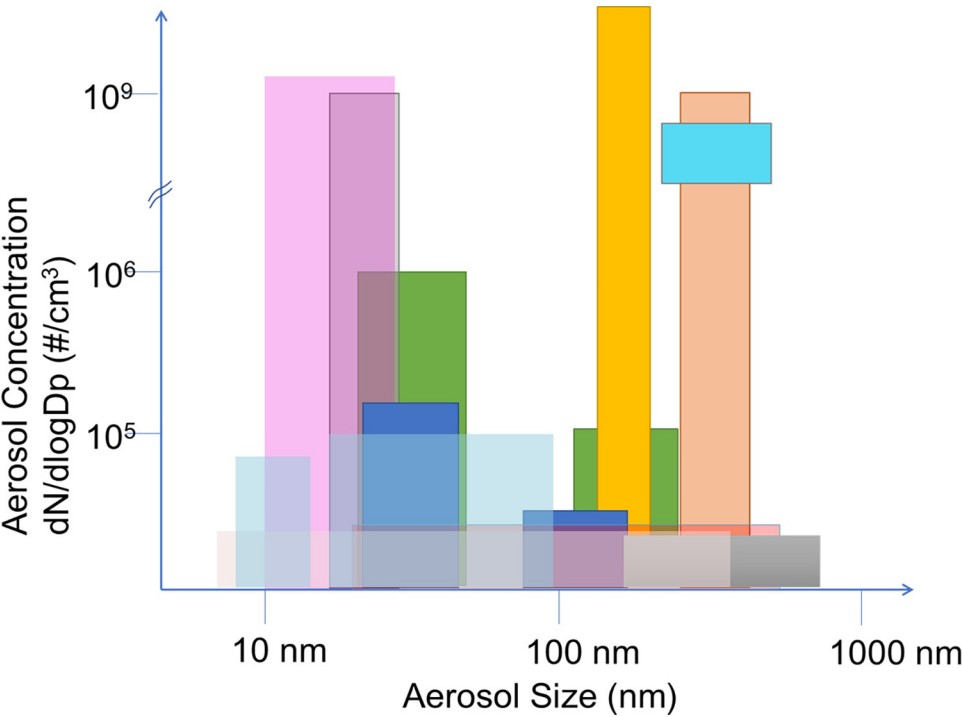

**Fig 1. Qualitative representation of the particle size versus concentration as reported by various studies (2013–2019).** Each colored box is representative of one reference. The variation in the results may be due to differences in experimental design (e.g., smoking machines, smoking topography, aerosol analysis equipment, and dilution factors).

accurate and repeatable characterization of ENDS aerosols. In particular, the aging of ENDS aerosols may lead to experimental artifacts in the detection of ultra-fine particles due to the dynamic nature of ENDS aerosols.

Given the possible negative health outcomes related to the inhalation of ENDS aerosols and ultra-fine particles: these particles may penetrate deeper into the lung, have cardiovascular effects [27] and can produce disproportionally severe health outcomes—combined with the limited research performed on ENDS operating and design parameters, we herein develop a unique instrumentation method that reduces aerosol aging, which is applied to systematically investigate the effects of atomizer/coil design, heating coil temperature, and PG:VG ratios on the aerosol characteristics within a single experimental setup. Our novel approach to reduce aerosol aging includes: a) positive pressure ENDS activation and sample collection, which to the authors' knowledge is not found in previously published work, b) minimization of both sample tubing length and dilution factors, and c) a high-resolution electrical low pressure impactor capable of simultaneous measuring of particle count data and particle mass, which is rarely featured in ENDS studies. These pronounced measures were taken to minimize experimental artifacts and accurately characterize ENDS aerosols including aerosol mass generation, particle count generation, and the particle mass and count size distributions for high wattage "sub-ohm" ENDS (i.e., having a coil resistance of less than 1 ohm).

## Materials and methods

### Atomizers and modules

A Reuleaux module ENDS (Wismec, Shenzhen, China), henceforth referred to as the "mod", was used in this investigation. The mod can deliver up to 200 W of power and has a

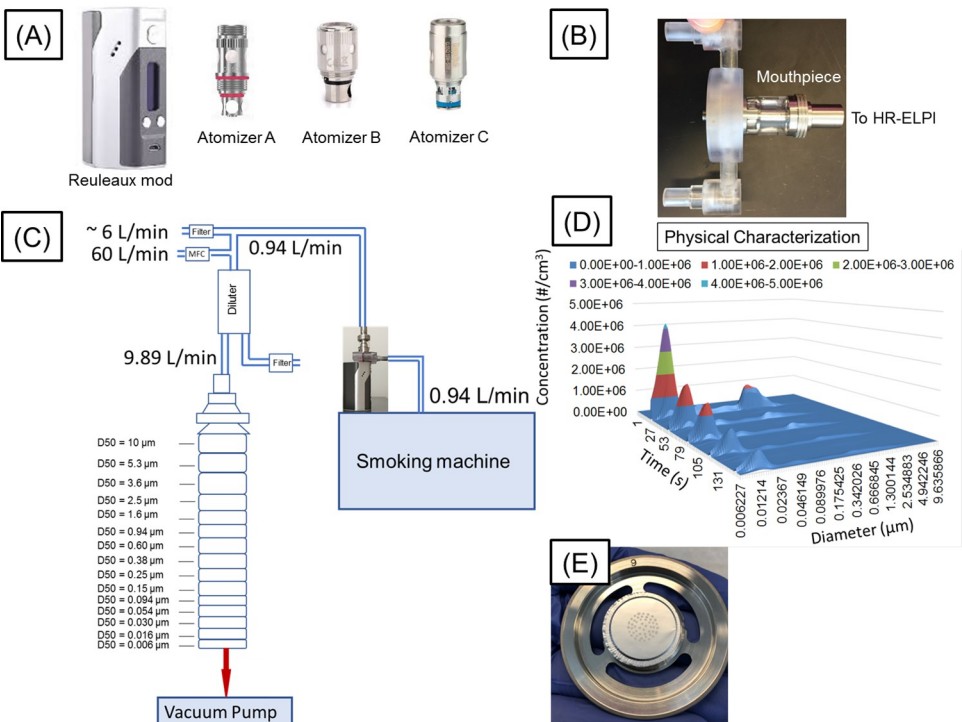

**Fig 2. Materials and experimental apparatus.** (A) The mod and the three atomizer coils chosen for this study (B) Modification at the mouthpiece of an atomizer for positive pressure operation of the smoking machine (air pushed from the smoking machine to the atomizer instead of pulled from the atomizer to the smoking machine) (C) Schematic of the flow loop built for the experiments (D) Size distributions obtained for an Atomizer plotted across time and over the entire size regime at 200˚C. Only the first five scans have been shown. The scans were obtained every 30 seconds. With each subsequent scan after the first one, the peak concentration appears to reduce. The color legend represents different ranges of concentration. For simplicity the subsequent plots shown in this manuscript have been averaged (as and when needed) and hence are not presented in the same format. (E) One stage of the ELPI shown with aluminum foil with deposits of the e-liquid in the middle. Gravimetric analysis was performed on these foils.

temperature control mode that is controlled using a DNA 200 microprocessor (Fig 2A) (Evolv LLC, Hudson, OH). The temperature was set at either 200˚C or 300˚C, consistent with our prior study [28]. Three different atomizers, designated atomizers A, B, and C for this study (Fig 2A), were the same types that were used previously. Each of the three atomizers has a Nickel 200 (99.6% pure wrought nickel) coil with a resistance of 0.15 Ω, but differed in internal air flow path length and cross-sectional area. All three atomizers are vertical coils that wrapped by the wick. Atomizer B is a dual coil, while atomizers A and C are single coil. Further details about the atomizers can be found in the prior study [28]. The mod was set to 100 W during all experimental test runs.

Temperature measurements were also kept the same as established in previous study (Dibaji, et al., 2018). The ENDS control circuitry measures the voltage, resistance and current as a function of time during operation. The temperature is determined by the resistance variation with temperature. The applied ENDS in this study utilizes the DNA 200 microchip (Evolv, LLC) a popular microprocessor that can measure and control temperature. The temperature control mode allows the user to specify the maximum coil temperature (100–300 ˚C) during operation. The accuracy of the DNA 200 in measuring and controlling the temperature was evaluated against thermocouple measurements in our previous study [28]. In that study, the power and temperature control were set at 100 W, and 300 ˚C, respectively. Thermocouples (type K) were secured to the top and middle sections of the heating coil using thermally

conductive cement and the coil temperature was recorded by thermocouples as well as the DNA 200 associated software (EScribe) during simulated puffing conditions using e-liquid composition of 65% VG, 35% PG, and 3 mg/mL of nicotine. Although thermocouple measurements showed variation in temperature along the length of the coil, there was an acceptable agreement between the average temperature recorded by thermocouples and EScribe software.

### Preparation of e-liquid

For the main e-liquid components, United States Pharmacopeia Grade (USP-grade) PG, USP-grade VG and 99% pure nicotine measured via gas chromatograph (GC-grade nicotine) (Sigma-Aldrich, St. Louis, MO) were used. PG and VG were mixed in volumetric proportions using 5 mL or 10 mL pipettors (Gilson, Middleton, WI) in clean 15 mL polypropylene centrifuge tubes (Corning Inc., Corning, NY). No further additives, such as flavors, were used. Amounts were verified by gravimetry using an analytical balance (Model AB54-S Mettler Toledo, Columbus, OH) with an accuracy of ± 0.4 mg. Nicotine concentration was 0.30 mg/mL.

### Smoking machine and its operation

A modified LX1E smoking machine (Borgwaldt, Hamburg, Germany), which has a single module to control puffs and a single linear port for holding the ENDS, was used in the study. A square puff profile with a puff volume of 55 mL per 3.5 seconds was programmed into the instrument. Since the study was designed to examine minimally aged aerosols, the LX1E machine was customized by the manufacturer to operate in a positive pressure mode. This ensured that the puffs generated by the ENDS would be pushed out and directly moved to the physical analyzer for characterization, as opposed to a single- or double-stroke piston-type sampling system, which impacts aerosol aging [18]. To operate in the positive pressure mode, special adapters were made for each atomizer, so the tubing from the modified LX1E would connect to the flow path of the atomizer correctly (Fig 2B). The schematic of the experimental set up is shown in Fig 2C.

### Physical and gravimetric characterization

The aerosols from the mouthpiece of the ENDS were directed towards a HR-ELPI (Dekati Ltd., Kangasala, Finland) for physical characterization. The HR-ELPI has 14 impactor stages and the 100 channel electrometer data is inverted using known impactor kernel functions preloaded in the manufacturer's software to obtain the size distribution [29]. Because the preliminary results suggested that the aerosols being examined from the ENDS were too concentrated for analysis using the HR-ELPI, the aerosols were diluted with room air at the HR-ELPI inlet as shown in Fig 2C. In this regard, the maximum concentration limit for the HR-ELPI is size dependent and is typically $2\times10^5$ #/cm3, $3.7\times10^6$ #/cm3 and $7.9\times10^7$ #/cm3 at approximately 10 μm, 0.1 μm, and 10 nm, respectively. The total dilution of the e-cigarette smoke is given by (60 + 6 + 0.94)/0.94 i.e. 71.2 where two sources of 6 L/minute and 60 L/minute enter the Dekati diluter (Dekati Ltd., Kangasala, Finland) and mix with the e-cigarette flow rate of 0.94 L/minute. Subsequently 9.89 L/minute of that total flow is sampled by the HR-ELPI and the rest is discarded. The HR-ELPI was operated at a scan rate of 1 Hz. The aerosol concentration data was recorded as a function of aerosol size. Note that although the puff was 3.5 seconds long, the HR-ELPI was found to detect a large concentration of aerosol for roughly 10 seconds for every puff before returning to baseline concentration. Because this temporal diffusion of the aerosols was induced by the instrument, the concentration across the full 10 seconds was summed for determining the maximum concentration per puff. The maximum concentration

was seen during the first puff 77% of the time and occurred within the first 3 puffs 94% of the time. The variation in the concentration of the aerosols with time was likely reflection of the intrinsic variation in the aerosol generation in the e-cigarettes. The average of three test runs (n = 3), are reported in the results section, unless the instrument was saturated with e-liquid and unable to return valid results. Concentration data collected for atomizer B at a PG:VG ratio of 50:50 at 200˚C is omitted entirely due to the impactor being overwhelmed.

An orthogonal approach of gravimetric analysis of the collected aerosol was implemented, by retrieving and weighing each impactor stage after physical characterization. This approach does not have the limitations described above for concentration measurements and thus all gravimetric data is reported in triplicates. Aluminum foils were employed that were pre-weighed and post-weighed for all 14 stages using a micro-balance (Model ME5-F, Sartorius, Goettingen, Germany) with readability of 1 µg (Fig 2E). The number of puffs collected were carefully and iteratively determined to be ten puffs at 200˚C and five puffs at 300˚C, respectively, with 30 seconds interval between each puff. Ten puffs at 200˚C enabled us to collect detectable amounts of nanoparticles, whereas at 300˚C we were limited to 5 puffs as collecting e-liquid from more puffs caused spilling of the e-liquid to subsequent stages thus impacting the results. The lowest non-zero standard deviation for the gravimetric measurements ($\sigma_g$) was 1 µg. Assuming the limit of quantitation (LOQ) is $10 \times \sigma_g$ [30], the minimum weight above which we can infer trends was established as 10 µg (i.e., 0.01 mg).

## Statistical analysis

A Student's t-test (two tailed, unequal variance) was performed using Microsoft Excel® (Office 365) to determine whether there were statistical differences among different temperatures, atomizers and PG:VG ratios ($p < 0.05$, $\alpha = 0.05$). A complete list of statistically significant results can be found in Table 1 in S1 Appendix.

## Results

Consistent with literature [31, 32], we analyzed the particle size data by grouping into three particle size categories: (1) less than 0.1 micron (i.e., ultra-fine particles), (2) 0.1 to 1 micron (range for fine particulate matter), and (3) greater than 1 micron (typically, coarse particulate matter). Accordingly, we report particle size based results, including concentration and mass, across these three aerosol size groups in Figs 3–6 for combinations of different atomizers, PG:VG ratios, and temperature.

### Characterization of ultra-fine particles

We first analyzed the aerosol particle concentrations, averaged across all puffs, for different atomizers and PG:VG ratios at 200˚C (Figs 3 and 4). Fig 3 reveals that high concentrations of ultra-fine particles were generated, particularly for atomizer A, although there was high sampling variability for this size group. Overall, atomizer A consistently produced the highest concentration of ultra-fine particles, with up to $2.2 \times 10^8$ particles/cm$^3$ generated at 200˚C (Fig 3), while ultra-fine particles generally accounted for the highest percentage of all particles, with the exception of atomizer C at the 65:35 PG:VG ratio (Fig 4).

Atomizer A generally showed higher particle concentration of ultra-fine particles than atomizer B under similar conditions, with statistical significance only at the 35:65 PG:VG ratio/300˚C (p = 0.002), while atomizer B generally showed higher particle concentration of ultra-fine particles than atomizer C, with statistical significance only at the 65:35 PG:VG ratio/200˚C (p = 0.03) (Fig 3). Moreover, higher temperatures (300˚C compared to 200˚C) resulted in higher particle concentration of ultra-fine particles, only at the 65:35 PG:VG ratio for

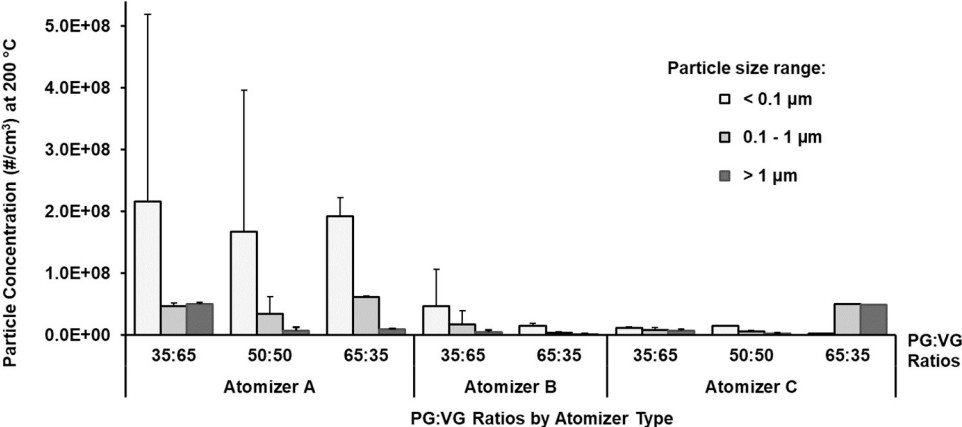

**Fig 3. Variability of aerosol concentration.** Test conditions at 200˚C for each size group across different brands of atomizer at varying PG:VG ratio.

atomizer C (p = 0.021). Aside from the specific conditions above, statistically significant correlations for independent parameters (atomizer, PG:VG ratio, or temperature) could not be drawn due to the high sampling variability as well as variability across different atomizers.

Next, we performed gravimetric analysis of the collected aerosols to study aerosol mass generation as a function of atomizer, temperature, and PG:VG ratio (Fig 5), confirming some of the general trends for ultra-fine particles in the aerosol concentration studies. Similar to the aerosol concentration results, atomizer A generally showed higher aerosol mass of ultra-fine particles than atomizer B, with statistical significance only at the 65:35 PG:VG ratio/200˚C (p = 0.049) (Table 1 in S1 Appendix). Atomizer C showed higher aerosol mass of ultra-fine particles than atomizer B, with statistical significance only at the 65:35 PG:VG ratio/300˚C (p = 0.004) (Table 1 in S1 Appendix). Similar to the aerosol concentration results, a higher temperature (300˚C compared to 200˚C) resulted in higher aerosol mass of ultra-fine particles only at the 65:35 PG:VG ratio for atomizer C (p = 0.020) (Table 1 in S1 Appendix). Unlike the

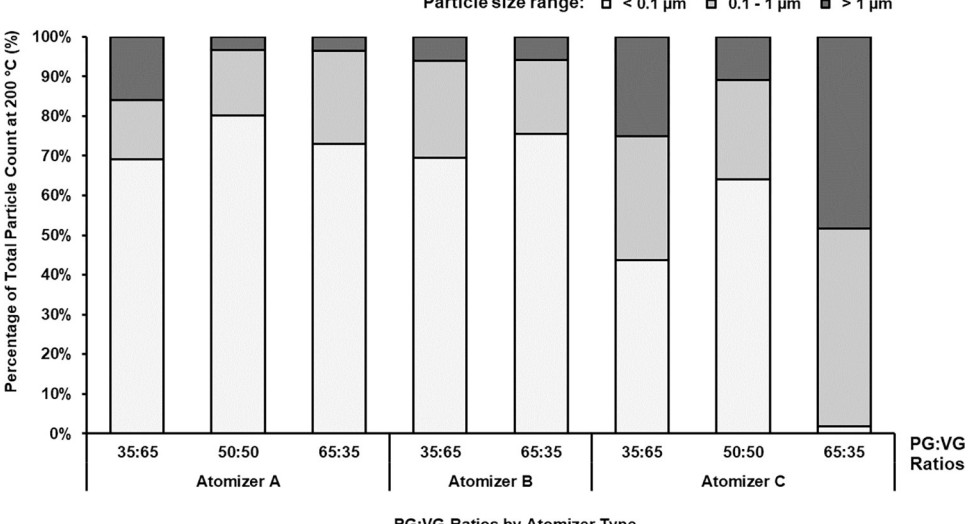

**Fig 4. Average aerosol concentration.** Test conditions at 200˚C for each size group, as a percentage of total particle concentration for that test condition, across different brands of atomizer at varying PG:VG ratio.

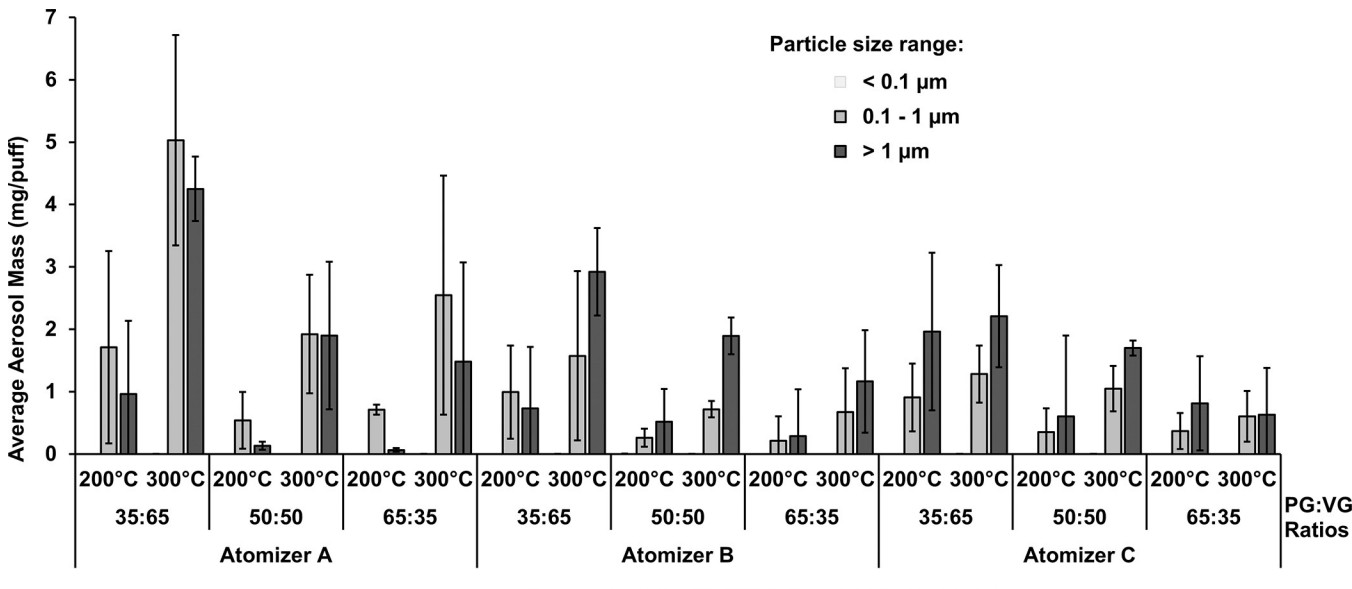

**Fig 5. Variability of mass of e-liquid.** Collected at 200˚C and 300˚C for ultra-fine, fine and coarse size group across different brands of atomizer and PG:VG ratio. The ultra-fine particles are not visually apparent as the amount of mass of these particles was minimal relative to the other two groups.

particle concentration data in Figs 3 and 4, the total particle mass from ultra-fine particles (< 0.1 μm) was minimal compared to fine particulate matter (0.1 to 1 μm) and coarse particulate matter (> 1 μm) (Fig 5).

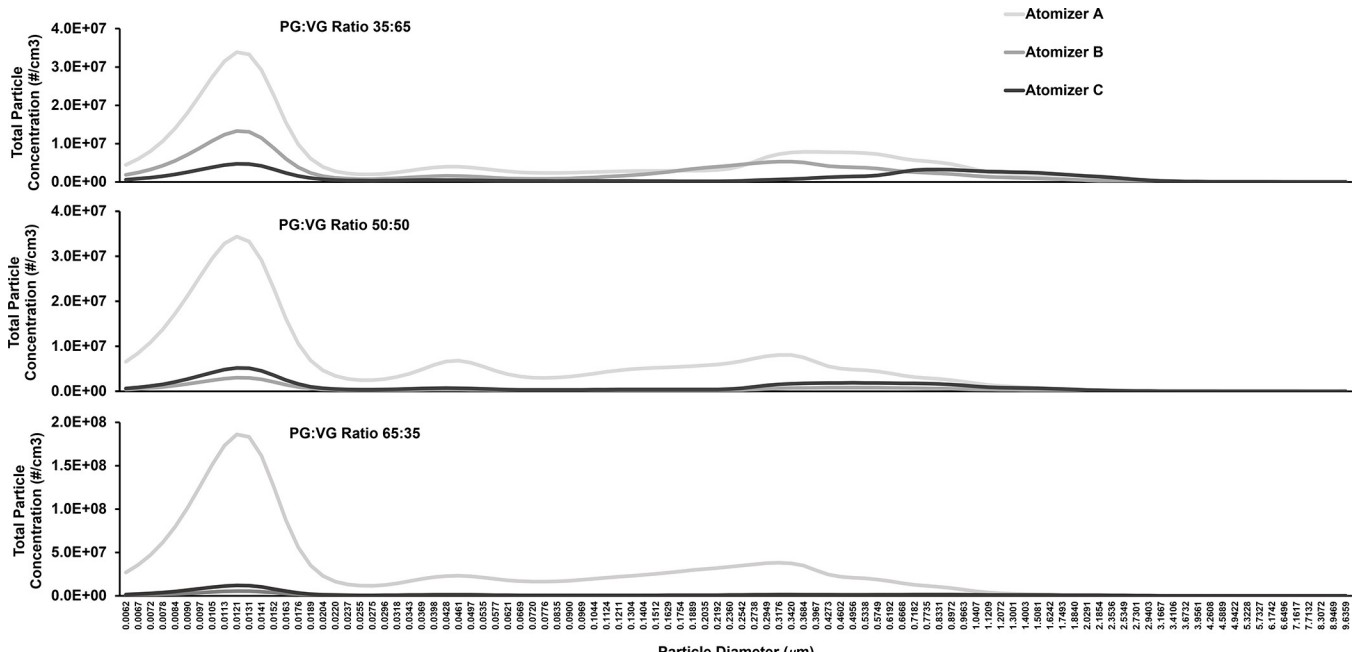

**Fig 6. Particle count size distribution.** Data collected for each atomizer and PG:VG ratio, from top to bottom 35:65, 50:50, and 65:35, at a coil temperature of 200˚C. Data is calculated from the sum of all particle counts for each time step, averaged for all test runs and puffs.

## Characterization of fine particles

Similar to the ultra-fine particle characterization above, atomizer A generally showed higher particle concentration of fine particles than atomizer B, with statistical significance only at the 65:30 PG:VG ratio for both temperatures ($p < 0.05$). Atomizer A also showed higher particle concentration of fine particles than atomizer C, with statistical significance only at the 35:65 PG:VG ratio/200˚C and at the 65:35 PG:VG ratio/300˚C ($p < 0.05$) (Fig 3). Higher PG content (65:35 PG:VG ratio compared to 50:50 and 35:65 ratios) generally resulted in higher particle concentration of fine particles, with statistical significance for atomizer A/300˚C and for atomizer C/200˚C ($p < 0.05$) (Fig 3).

Next, gravimetric analysis of the collected aerosols confirmed some of the general trends for fine particles in the aerosol concentration studies (Fig 5). Similar to the aerosol concentration results, atomizer A generally showed higher aerosol mass of fine particles than that of atomizer B and atomizer C, with statistical significance at the 35:65 PG:VG ratio/300˚C and at the 50:50 PG:VG ratio/300˚C ($p < 0.05$). Reflecting the trends for ultra-fine particles, higher temperatures (300˚C compared to at 200˚C) resulted in higher aerosol mass of fine particles, with statistical significance for atomizer A at the 35:65 and 50:50 PG:VG ratios, and for atomizer B at the 50:50 PG:VG ratio ($p < 0.05$). Lower PG content (35:65 PG:VG ratio compared to 50:50 and 65:35 ratios) generally resulted in higher aerosol mass of fine particles, with statistical significance for atomizer A at 300˚C ($p < 0.05$) (Fig 5).

## Characterization of coarse particles

For atomizer A, lower PG content (35:65 PG:VG ratio compared to 50:50 and 65:35 ratios) resulted in higher particle concentration of coarse particles, with statistical significance only at 200˚C ($p < 0.05$) (Fig 3). However, for atomizer C, higher PG content (65:35 PG:VG ratio compared to 50:50 and 35:65 ratios) resulted in higher particle concentration of coarse particles, with statistical significance only at 200˚C ($p < 0.05$) (Fig 3). Clear trends could not be identified for different atomizers for aerosol concentrations of coarse particles.

Gravimetric analysis of the collected aerosols of the coarse particles showed that atomizer A resulted in higher aerosol mass of coarse particles than that of atomizer C, with statistical significance only at the 35:65 PG:VG ratio/300˚C. Reflecting the trends for ultra-fine and fine particles, higher temperatures (300˚C compared to at 200˚C) resulted in higher aerosol mass of coarse particles, with statistical significance for atomizer A at the 35:65 PG:VG ratio, for atomizer B at the 35:65 PG:VG and 50:50 PG:VG ratios ($p < 0.05$). Lower PG content (50:50 PG:VG ratio compared to the 65:35 ratio) generally resulted in higher aerosol mass of coarse particles, with statistical significance for atomizer A at 200˚C ($p < 0.035$) (Fig 3).

## Discussion

Both the size of the particles as well as their mass dictate the effects of the aerosols generated by ENDS on human health. The smallest nanoparticles (0.01–0.02 μm) get primarily deposited in the airways by diffusion and may get cleared by mucocilliary action, but they can also be absorbed by blood and then translocate to the brain [33]. Inhalation of approximately 0.020 μm nanoparticles may result in increased pulmonary inflammatory response [34]. Increased alveolar inflammation induced by nanoparticles may not only lead to worsening of lung disease but also an increased risk of cardiovascular disease [35]. Furthermore, nanoparticles have increased surface-area-to-mass ratios, giving them higher surface reactivity, resulting in more biologically active particles. Particles between 0.020 μm and 0.250 μm in size are retained more readily in the interstitial lung tissues [36, 37]. A significant proportion of the larger submicron aerosols [38, 39] can reach the alveoli and cross over to the systemic

circulation. In fact, through numerical computations, ENDS submicron aerosols have been shown to deposit at approximately twice the density in the lung's pulmonary regions as conventional combusted cigarette smoke [25]. Particles larger than 1 µm are also filtered out mostly in the extrathoracic region by impaction but, given their size, their mass fraction is the largest. Additionally, different PG:VG ratios, in and of themselves, have been shown to have negative impact on lung function in mice [40], as they can affect the particle size distribution of the resulting aerosol thereby impacting deposition patterns in the lung.

Based on the rationale above, we herein designed a study that varies coil temperature, PG:VG ratio, and coil design to evaluate aerosol concentration and aerosol mass with respect to three particle size categories: ultra-fine particles ($< 0.1$ µm), fine particulate matter (0.1–1 µm), and coarse particulate matter ($> 1$ µm). Our objective was to better characterize the generation of ultra-fine particles in ENDS aerosols, which is currently limited by experimental artifacts during measurement due to the dynamic nature of ultra-fine particles and aging of aerosols.

Aerosol particle concentrations produced at 200˚C from various combinations of atomizer design and PG:VG ratios are presented in Fig 3. Atomizer A generally produced one to two orders of magnitude more ultra-fine particles than did atomizers B and C, for all PG:VG ratios. Atomizer A was also longer than the other two atomizers which may resulted in a greater surface contact between the heating coil and the wick/e-liquid resulting in more efficient heat transfer and higher particle generation. Further, for all atomizers and PG:VG ratios (except for atomizer C at 65:35), the ultra-fine particle size group had the greatest number of particles. Our result showing higher concentrations of ultra-fine particles than those of fine or coarse sized particles is also supported by additional work [39, 41–43]. To further identify trends in the particle count data, we performed a relative comparison among different particle size groups, as a percentage of the total particle count, in Fig 4 based on the data presented in Fig 3. Ultra-fine particles generally constituted more than 50% of the total concentration of all particles generated at 200˚C (Fig 4). However, the total aerosol mass of ultra-fine particles was minimal compared to the aerosol masses of fine and coarse particles, and showed high sample variability (Fig 4) likely because the measurement of the total aerosol mass of ultra-fine particles for this experiment was lower than the instrument defined LOQ (0.01 mg). Thus, for the ultra-fine particle size group, the particle counts or concentration (Fig 3) may provide more reliable results than the aerosol mass in analyzing the effects of various experimental parameters.

We next conducted gravimetric measurements of aerosols at the three size groups produced by ENDS (Fig 5), which showed positive correlations between temperature and overall aerosol mass for the fine and coarse particle size groups. Note that the values experimentally obtained have then been corrected for the dilution that the e-cigarette puff undergoes in the process of going through the diluter and because the ELPI only samples part of the diluted air. The mean (+/- s.d.) ratio of aerosol mass generated at 300˚C compared to 200˚C on a per puff basis (mg/puff) is 2.70 ± 2.67, averaged across all atomizers and PG:VG ratios. For atomizers A and B, the amount of overall aerosol mass in the fine and coarse particle size groups generally increased by a factor of 3 to 5 with an increase in temperature across different PG:VG ratios. With increased temperatures (300˚C compared to 200˚C), statistically significant increases in the aerosol mass of fine and coarse particles were identified for atomizer A and B at various PG:VG conditions ($p < 0.04$). For atomizer C, no statistically significant changes in aerosol mass for the fine and coarse particle size groups were observed between the two temperatures. Moreover, the temperature effects on aerosol mass generation were more pronounced for the coarse particle size group compared to the fine particle size group, for atomizers A and B (and for C at the 50:50 PG:VG ratio).

The increase in the overall aerosol mass of coarse particulate matter at higher temperatures is likely due to the increase in total aerosol mass generated during puffing, which generates a higher number of primary particles and aerosol mass available for growth through agglomeration and condensation [41]. This association between temperature and aerosol particle size distribution, demonstrated with various coil designs and PG:VG ratios herein, indicates the importance of temperature in the safety and performance of ENDS. Our results correspond well with previous studies [14, 41]. This trend of higher production of coarse particles at higher temperatures could not be identified for Atomizer C, likely due to the poor temperature control exhibited by this atomizer when used in conjunction with the Reuleaux mod, as previously demonstrated [28].

In examining the impact of PG:VG ratios, our results showed lower PG content generally resulted in higher aerosol mass of fine and coarse particles. However, measurements using particle count and concentration showed mixed results, depending on the atomizer and the particle size group. Thus, the methods of analyses (aerosol concentration versus aerosol mass) and the coil design may affect the interaction between PG:VG ratios and particle size distribution. Indeed, Pourchez et al. [14] found that increased VG content resulted in increased overall aerosol mass generation while, conversely, Baassiri et al. [15] found an increase in PG content resulted in increased aerosol mass generation. Our findings reported in Fig 5 are consistent with the work of Pourchez et al. [14] since mass/puff generally increased with decreasing PG content for most atomizers at the given temperatures. Pourchez et al. [14] also established a correlation between an increase in aerosol mass generation and power output of the ENDS. Moreover, Pourchez et al. [14] and Baassiri et al. [15] both reported that a greater PG content generates smaller particles at lower power settings than from a VG dominant e-liquid. Furthermore, consistent with prior studies such as Floyd et al., 2018, we also observed a multi-modal size distribution. For atomizer A peaks appeared around 12 nm, 40 nm, and 200–300 nm across all PG:VG ratios (Fig 6). For atomizers B and C a bimodal distribution was observed with peaks appearing around 12 nm, and then at > 500 nm.

Overall, the aerosol concentration studies were limited by high sampling variability as well as variability across different atomizers, precluding us from making general conclusions regarding the effects of various parameters on ENDS aerosols. However, the gravimetric studies showed that lower temperatures (200˚C), and higher PG:VG ratios at 300˚C yielded the least number of ENDS aerosol particles. Still, given the diversity of aerosol measurements and correlations in literature, there has yet to be a clear consensus and understanding of various engineering and chemical variables of ENDS and implications on the aerosols generated. Thus, additional studies that control for aerosol aging and minimize experimental artifacts can confirm the general findings herein.

The next important aerosol characteristic is the aerosol mass, which correlates with the dose. Several studies have pointed out the cytotoxicity of ENDS aerosols and the addiction properties of nicotine [44]. While multiple studies suggested that the ENDS are less toxic than conventional cigarettes, the popularity of ENDS is primarily in the youth population [45, 46]. It is noteworthy that the total surface area of the lungs is significantly less in younger populations compared to adults [47] implying a significantly higher dose per unit area for young users. In fact, systemic toxic effects have been reported in infants *in utero* [48]. Though the aerosol mass generation is investigated here, studies of possible toxic effects on youth from aerosol mass distributions may be warranted due to youth use patterns [49], and physical lung differences, combined with the harmful effects of the flavors in ENDS (which were outside the scope of this study).

## Limitations of the study

We note several limitations in our study herein. First, we performed resistance-based temperature measurements for the ENDS rather than measuring the actual temperature of the coil. However, our previous work demonstrated that the actual temperature can be significantly different from the resistance-based measurement depending on the coil design [28]. We did not correct for this source of error in this study. Second, although we selected 3.5 second puffs in the smoking machine, the actual aerosol concentration in the HR-ELPI was elevated for approximately 10 seconds for every puff. We corrected for this elevation by integrating and reporting the aerosol concentrations within these 10 seconds and then averaging over 10 puffs at 200˚C in Fig 3. Still, the aerosols may diffuse during the process of transport from the ENDS to the HR-ELPI. Other factors that may have contributed towards aging of aerosols were the tubing connecting the ENDS mouthpiece to the HR-ELPI diluter, the overwhelming of HR-ELPI under some conditions, and the dilution of aerosols. However, we do not expect the above factors to have strong influences on the aging of aerosols, since we found high concentrations of particles smaller than 0.1 micron. Another related aspect was evaporation. Since PG is significantly more volatile than VG there can be a preferential tendency of PG to evaporate. Given the low surface area of the collected samples in the HR-ELPI plates, the high boiling point for PG (188.2˚C) and VG (290˚C), because the collection temperature was close to ambient is it unlikely that the evaporation of PG was significantly higher compared to glycerol after getting collected at the impactor plates. However, further systematic studies may be required to address this issue. In addition, as noted before, the 3.5 s puff appears to be diffused into a 10 s puff when analyzed by the HR-ELPI. It is possible that the increased residence time at the diluter may be also cause additional evaporation of the aerosols, and preferential evaporation of PG. We also recommend future studies consider orthogonal validation of gravimetric analysis using filter pads which was beyond the scope of the current study. Lastly, the effects of nicotine concentration, flavors and the chemical compositions of the ENDS aerosols were outside the scope of this study, but further studies to examine the impact of such factors on aerosol characteristics will be informative.

## Conclusions

An added layer of complexity that is often overlooked with electronic cigarette aerosols is that their properties are likely to be very dynamic, with hygroscopicity and coagulation [50] impacting deposition in the lungs, making it important to characterize ENDS aerosols with instruments that can collect data rapidly (~ 1 second /scan). Indeed, a survey of literature (Fig 1) combined with our own findings reveal that only rapid analyzers such as the ELPI, FMPS and DMS500 are sufficiently sensitive in detecting the sub 0.1 μm particles. Some have debated whether the presence of nanoparticles is an artifact of the measurement methods used [9, 21, 51]. However, the presence of nanoparticles in ENDS aerosols [5, 18, 39] is supported in theory by numerous works demonstrating that nanoparticles have been generated via resistively heated wire techniques [8]. The role ultra-fine particles play in toxicity combined with variability of measurement techniques underscores the importance of using rapid/real-time analyzers to characterize aerosols from ENDS.

To reduce the experimental artifacts associated with ENDS aerosol aging and accurately characterize ultra-fine particles, we designed a unique instrumentation method combining positive pressure sample collection, minimization of sample tubing length and dilution factors, and a HR-ELPI. We applied this method to investigate the effects of coil design, coil temperature, and PG:VG ratios on the aerosol mass generation, particle count generation, and particle mass and count size distributions for a high-powered ENDS. Our studies highlight the

importance of orthogonal analysis, such as the combination of aerosol particle concentration ($\#/cm^3$) data and average aerosol mass (mg/puff) data for each of the ultra-fine, fine, and coarse particle size groups, which allowed trends to be confirmed or further scrutinized. For some interactions (e.g., PG:VG ratios and particle size distribution), the method of analyses (aerosol concentration versus aerosol mass) appeared to have an impact on the trend direction.

Our results indicate an increase in the overall aerosol mass of fine and coarse particles when the e-liquid contains higher VG content and lower PG content. Moreover, we identified an increase in the overall aerosol mass with a higher temperature, likely due to a higher number of primary particles and aerosol mass available for agglomeration, indicating the importance of coil temperature in the health risks posed by, and performance of, ENDS. Important to health outcomes, our results show a high concentration of ultra-fine particles from some atomizers and coil temperatures relative to the fine and coarse particle size groups, which may warrant further research into identifying ENDS design parameters that most impact ultra-fine particle generation. Overall, this work provides a novel method of characterizing ENDS aerosols to minimize aerosol aging and provides a better understanding on how various ENDS design parameters affect aerosol characteristics.

## Supporting information

**S1 Appendix. Student's t-test results and count median diameter analysis.**
(DOCX)

**S2 Appendix. Raw particle mass and count data.**
(XLSM)

## Acknowledgments

**Disclaimer:** The findings and conclusions in this report are those of the authors and do not necessarily represent the official position of the Food and Drug Administration.

## Author Contributions

**Conceptualization:** Berk Oktem, Todd Cecil, Suvajyoti Guha.

**Data curation:** Suvajyoti Guha.

**Funding acquisition:** Todd Cecil.

**Investigation:** Seyed Ahmad Reza Dibaji, Berk Oktem.

**Methodology:** Seyed Ahmad Reza Dibaji, Berk Oktem, Todd Cecil, Samanthi Wickramasekara, Suvajyoti Guha.

**Project administration:** Samanthi Wickramasekara, Matthew Myers, Suvajyoti Guha.

**Supervision:** Matthew Myers, Suvajyoti Guha.

**Validation:** Seyed Ahmad Reza Dibaji, Berk Oktem.

**Visualization:** Seyed Ahmad Reza Dibaji.

**Writing – original draft:** Lee Williamson, Suvajyoti Guha.

**Writing – review & editing:** Seyed Ahmad Reza Dibaji, Berk Oktem, Jenna DuMond, Jimin P. Kim, Samanthi Wickramasekara, Matthew Myers.

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
