## [Decision Letter · Decision Letter 0]

15 Jul 2022

PONE-D-21-27691

Title -  Characterization of Aerosols Generated by High-Power Electronic Nicotine Delivery Systems (ENDS): Influence of Atomizer, Temperature and PG:VG Ratios

PLOS ONE

Dear Dr. Williamson,

Thank you for submitting your manuscript to PLOS ONE. After careful consideration, we feel that it has merit but does not fully meet PLOS ONE’s publication criteria as it currently stands. Therefore, we invite you to submit a revised version of the manuscript that addresses the points raised during the review process.

We look forward to receiving your revised manuscript.

Kind regards,

Ali Sharif, Ph.D.

Academic Editor

PLOS ONE

Journal Requirements:

4. Please amend the manuscript submission data (via Edit Submission) to include author Lee Williamson.

5. Please amend your authorship list in your manuscript file to include author Raymond Williamson.

Reviewers' comments:

Reviewer's Responses to Questions

**Comments to the Author**

1. Is the manuscript technically sound, and do the data support the conclusions?

Reviewer #1: Yes

Reviewer #2: Yes

2. Has the statistical analysis been performed appropriately and rigorously? 

Reviewer #1: Yes

Reviewer #2: Yes

3. Have the authors made all data underlying the findings in their manuscript fully available?

Reviewer #1: Yes

Reviewer #2: Yes

4. Is the manuscript presented in an intelligible fashion and written in standard English?

Reviewer #1: Yes

Reviewer #2: Yes

5. Review Comments to the Author

Reviewer #1: *Please add information to support why you chose the concentration of nicotine being delivered by the ENDS to be less than that delivered by ENDS used publicly.

*Provide a reason for using GC grade nicotine. Is this grade of nicotine used in regular marketed ENDS? Are the two types of nicotine comparable and do they generate similar results?

*Finally I think there should have been a commonly used ENDS product and brand of nicotine included in the study.

Reviewer #2: It is an interesting study that demonstrates the factors contributing to the electronic nicotine delivery systems. This manuscript can be accepted following some revisions.

Describe in short the statistical analysis carried out on Figure 3,4, 5 and 6. Also give some statistical significance symbols on these images.

6. PLOS authors have the option to publish the peer review history of their article (what does this mean?). If published, this will include your full peer review and any attached files.

Reviewer #1: **Yes: **Zakiah Zeb

Reviewer #2: **Yes: **Muhammad Furqan Akhtar

---

## [Author Response · Author response to Decision Letter 0]

9 Nov 2022

Revisions submitted as requested in the "Decision Letter".

---

## [Editor Report · Decision Letter 1]

6 Dec 2022

Title -  Characterization of Aerosols Generated by High-Power Electronic Nicotine Delivery Systems (ENDS): Influence of Atomizer, Temperature and PG:VG Ratios

PONE-D-21-27691R1

Dear Dr. Williamson,

We’re pleased to inform you that your manuscript has been judged scientifically suitable for publication and will be formally accepted for publication once it meets all outstanding technical requirements.

Kind regards,

Ali Sharif, Ph.D.

Academic Editor

PLOS ONE

Additional Editor Comments (optional):

The authors have addressed the reviewer comments. The article can be processed for publication.
---

## [Editor Report · Acceptance letter]

8 Dec 2022

PONE-D-21-27691R1 

Characterization of Aerosols Generated by High-Power Electronic Nicotine Delivery Systems (ENDS): Influence of Atomizer, Temperature and PG:VG Ratios 

Dear Dr. Williamson:

I'm pleased to inform you that your manuscript has been deemed suitable for publication in PLOS ONE. Congratulations! Your manuscript is now with our production department. 

Kind regards, 

on behalf of

Dr. Ali Sharif 

Academic Editor

PLOS ONE